# Bile Acids as Emerging Players at the Intersection of Steatotic Liver Disease and Cardiovascular Diseases

**DOI:** 10.3390/biom14070841

**Published:** 2024-07-12

**Authors:** Josh Bilson, Eleonora Scorletti, Jonathan R. Swann, Christopher D. Byrne

**Affiliations:** 1School of Human Development and Health, Faculty of Medicine, University of Southampton, Southampton SO16 6YD, UK; eleonora.scorletti@pennmedicine.upenn.edu (E.S.);; 2National Institute for Health Research Southampton Biomedical Research Centre, University of Southampton, University Hospital Southampton National Health Service Foundation Trust, Southampton SO16 6YD, UK; 3Division of Genetics, Perelman School of Medicine, University of Pennsylvania, Philadelphia, PA 19104, USA

**Keywords:** metabolic dysfunction-associated steatotic liver disease, cardiovascular disease, cardiac disease, obeticholic acid, farnesoid X receptor, cardiac disease, resmetirom, bile acids, bile acid receptors

## Abstract

Affecting approximately 25% of the global population, steatotic liver disease (SLD) poses a significant health concern. SLD ranges from simple steatosis to metabolic dysfunction-associated steatohepatitis and fibrosis with a risk of severe liver complications such as cirrhosis and hepatocellular carcinoma. SLD is associated with obesity, atherogenic dyslipidaemia, and insulin resistance, increasing cardiovascular risks. As such, identifying SLD is vital for cardiovascular disease (CVD) prevention and treatment. Bile acids (BAs) have critical roles in lipid digestion and are signalling molecules regulating glucose and lipid metabolism and influencing gut microbiota balance. BAs have been identified as critical mediators in cardiovascular health, influencing vascular tone, cholesterol homeostasis, and inflammatory responses. The cardio-protective or harmful effects of BAs depend on their concentration and composition in circulation. The effects of certain BAs occur through the activation of a group of receptors, which reduce atherosclerosis and modulate cardiac functions. Thus, manipulating BA receptors could offer new avenues for treating not only liver diseases but also CVDs linked to metabolic dysfunctions. In conclusion, this review discusses the intricate interplay between BAs, metabolic pathways, and hepatic and extrahepatic diseases. We also highlight the necessity for further research to improve our understanding of how modifying BA characteristics affects or ameliorates disease.

## 1. Introduction

Steatotic liver disease (SLD) is a growing global health concern, affecting approximately 25% of the population worldwide and approximately 30% of individuals in the United States [1]. The prevalence is likely to be an underestimate due to the asymptomatic nature of SLD and the need for blood tests and imaging tests, or tissue biopsy, to confirm the diagnosis. SLD encompasses a spectrum of liver conditions, ranging from simple steatosis to metabolic dysfunction-associated steatohepatitis (MASH) and fibrosis. Simple steatosis can progress to MASH, characterized by hepatocyte ballooning and inflammation, with or without fibrosis. MASH affects approximately 1.5% to 6.45% of the general population and is a significant contributor to cirrhosis, hepatocellular carcinoma, and liver-related mortality [2]. The prevalence of MASH is projected to increase by 63% between 2015 and 2030, posing a substantial burden on global healthcare systems [3].

SLD is a multisystem disease, and it is associated with obesity, atherogenic dyslipidaemia, and insulin resistance [4]. These comorbidities contribute to the increased cardiovascular risk seen in SLD patients. While 10–25% of SLD patients may develop MASH, leading to severe liver complications, cardiovascular disease (CVD) remains the leading cause of mortality in this population [5,6,7]. Identifying SLD is therefore crucial for CVD prevention and treatment, necessitating increased clinician awareness.

Bile acids (BAs), known for their role in lipid digestion and absorption, also act as signalling molecules regulating glucose, lipid, and amino acid metabolism and maintaining gut microbiota homeostasis [8]. Elevated BA concentrations in liver disease can potentially harm the heart, and BAs have been recognized as key metabolic regulators with potential therapeutic implications for SLD and MASH [9]. Recent studies indicate that BAs and their derivatives can either promote or mitigate MASH, emphasizing the need for further research to understand these effects and develop reliable treatments based on BAs that also do not cause clinically significant side effects.

## 2. Bile Acid Synthesis, Regulation, and Key Signalling Pathways

The enterohepatic circulation of BAs involves the liver, biliary tract, intestine, portal venous circulation, colon, systemic circulation, and kidneys. Bile, an iso-osmotic micellar solution produced by the liver, contains bile acids, water, electrolytes, phosphatidylcholine, cholesterol, and bilirubin. BA synthesis is crucial for lipid digestion and absorption, cholesterol catabolism, fat-soluble vitamin absorption, and glucose and energy homeostasis.

Primary BAs are synthesized from cholesterol in hepatocytes via two pathways (Figure 1):The “classic” natural pathway (quantitatively more important) in which, firstly, cholesterol is converted by the enzyme cholesterol 7α-hydroxylase (CYP7A1) to 7α-hydroxycholesterol. Secondly, the enzyme 3β-hydroxy-Δ5-C27-steroid dehydrogenase/isomerase reduces 7α-hydroxycholesterol to 7α-hydroxy-4-cholesten-3-one. Thirdly, 7α-hydroxy-4-cholesten-3-one forms chenodeoxycholic acid (CDCA) or/and cholic acid (CA) through the action of sterol 12α-hydroxylase (CYP8B1).The “alternative” acidic pathway in which cholesterol is converted to 27-hydroxy-cholesterol by CYP27A1 and proceeds via 3β,7α-dihydroxy-5-cholestenoic acid to form CDCA [10].

Before secretion into the bile canaliculi, primary BAs are conjugated to taurine or glycine to enhance their hydrophilicity and acidic strength. Conjugated BAs facilitate lipid digestion and the absorption of lipid-soluble vitamins. In the intestine, BAs interact with gut microbiota in three instances: (i) bile acids are deconjugated by gut microbial bile salt hydrolases (BSHs) to form unconjugated bile acids, which can be absorbed and returned to the liver for reconjugation; (ii) a small portion of CDCA is modified through epimerization by gut microbiota to produce urodeoxycholic acid (UDCA); and (iii) in the colon, bacterial CYP7A1 converts CA to deoxycholic acid (DCA) and CDCA to lithocholic acid (LCA), reducing BA solubility. These secondary BAs are either absorbed and returned to the liver or excreted in the faeces.

Deconjugation and dehydroxylation of primary BAs is facilitated by several microbial species including those within the Lactobacillus, Enterococcus, Bifidobacterium, and Clostridium genera. Secondary BAs are metabolised by the liver and gut microbiota to form tertiary BAs through sulfation, hydroxylation, and glucuronidation [11]. LCA activates the pregnane X receptor (PXR), a nuclear receptor expressed primarily in the liver and in the intestine. PXR has an important role in the hepatic detoxification of toxic metabolites including unmodified LCA. The activation of PXR induces the sulfotransferase enzyme that is responsible for the sulfation of LCA; after sulfation or glucuronidation, LCA can enter the intestine and be rapidly excreted. These two reactions decrease the cytotoxic effect of LCA [12].

BAs are an important regulator of lipid homeostasis. They are natural ligands of the farnesoid X receptor (FXR), a nuclear receptor expressed in the liver, intestine, kidney, and adipose tissue. Bile acid-activated FXR indirectly represses (via the induction of the short heterodimer partner, SHP) sterol regulatory element-binding transcription factor 1 and carbohydrate response element binding protein, which is heavily involved in hepatic de novo lipogenesis. Moreover, BA-activated FXR activates peroxisome proliferator-activated receptor alpha, which in turn induces the expression of genes encoding the enzymes involved in the β-oxidation of fatty acids that inhibit triglyceride synthesis and very low-density lipoprotein export [13].

A diet rich in fats is likely to result in higher levels of liver cholesterol, leading to an increased synthesis of BAs. The consequences of this altered metabolism are the accumulation and impaired detoxification of secondary Bas, which are more hydrophobic and toxic than primary BAs.

Secondary BAs have the following toxic effects:Increased intestinal permeability with decreased expression of tight junctions. This allows for the translocation of endotoxin products (i.e., LPS from Gram-negative bacterial membrane) from the intestine to the bloodstream and thus to the portal vein and into the liver, causing inflammation and cytokine production [14].Apoptosis of enterocytes and hepatocytes. The hydrophobicity of secondary bile acids allows for their interaction with the phospholipids in the cell membranes of hepatocytes and enterocytes. Secondary bile acids can also travel to the cytosol inducing perturbations of the mitochondrial membrane. The alterations of the mitochondria stimulate the production of reactive oxygen species that, in turn, enhance mitochondrial permeability and cause the release of cytochrome c and other factors that form the apoptosome, ultimately causing cellular apoptosis and necrosis [15].

The complex interplay between BAs, the liver, gut microbiota, and various physiological processes underscores the critical role of BAs in maintaining lipid homeostasis and overall cardiometabolic health. The synthesis, modification, and detoxification of BAs are essential for lipid digestion, cholesterol regulation, and the absorption of fat-soluble vitamins. However, disruptions in BA metabolism, particularly the accumulation of toxic secondary BAs, can have detrimental effects, such as increased intestinal permeability, mitochondrial dysfunction, and cellular apoptosis. Understanding these processes is vital for developing therapeutic strategies to manage metabolic diseases and improve liver and gut health.

## 3. The Role of Trans-Genomic Bile Acid Metabolism in the Pathogenesis and Progression of MASLD

### 3.1. The Associations between Bile Acid Concentrations and MASLD Severity

As discussed by others [16,17,18], numerous studies have indicated that circulatory concentrations of BAs are altered in participants with obesity and obesity-associated metabolic diseases. Thus, it is perhaps not surprising that BA concentrations have also been shown to be modified in patients with SLD compared to healthy controls and associated with liver disease severity (Appendix A). Whilst discordance exists, previous observations generally indicate that the presence of SLD is associated with an increase in total BA concentrations (Appendix A). In 2023, Lai and colleagues conducted a meta-analysis of 19 studies exploring circulating BA concentrations in patients with metabolic dysfunction-associated SLD (MASLD) (*n* = 43,229) compared to healthy controls (*n* = 111,578) [19]. According to this work, total BA concentrations were shown to be different in patients with MASLD compared to healthy controls, with a total random-effects standard mean difference of 1.03 (95% CI: 0.63–1.42, *p* < 0.001) and substantial heterogeneity (*I*^2^ = 96%). When considering specific BA species, circulating concentrations of ursodeoxycholic acid (UDCA), taurocholic acid (TCA), CDCA, taurochenodeoxycholic acid (TCDCA), GCA, glycoursodeoxycholic acid (GUDCA), glycochenodeoxycholic acid (GCDCA), tauroursodeoxycholic acid (TUDCA), and CA were consistently higher in patients with MASLD compared to healthy controls [19]. Whilst differences in these BAs appeared to be more pronounced in patients with vs. without MASH, there was insufficient evidence to explore associations between BA concentrations and liver fibrosis severity [19]. Indeed, only a few studies have investigated the associations between circulatory BAs and liver fibrosis severity in patients with SLD (Appendix A) [20,21,22,23].

In the study by Adams et al., while total BA concentrations were associated with liver fibrosis severity in patients with SLD, after adjusting for potential confounding factors (age, sex, body mass index (BMI), hypertension, type 2 diabetes, total cholesterol, homeostatic model assessment for insulin resistance (HOMA-IR), high-density lipoprotein cholesterol (HDL-C), and triglyceride (TAG)), only GCA, GCDCA, and GDCA concentrations were higher in patients with SLD with more (F3/F4) vs. less (F0–2) severe liver fibrosis [20]. In support of this, Nimer and colleagues found that concentrations of GCA, GCDCA, GUDCA, and 7-Keto-DCA were positively associated with liver fibrosis severity in patients with SLD; however, the authors did not consider the impact of potential confounding factors [23]. Concentrations of 12α-OH BAs (sum of CA, DCA, TCA, GCA, TDCA, and GDCA) were found to be 14-fold higher in patients with SLD and fibrosis compared to healthy controls (healthy control concentrations were approximately 200 ng/mL); however, no adjustments for potential confounding factors were performed [24]. Work published in 2022 by Kasai and colleagues indicated that circulating concentrations of CA and LCA (predominantly unconjugated LCA) were associated with liver fibrosis severity in patients with SLD independently of BMI and HOMA-IR [21]. Whilst others have reported differences in secondary BAs in patients with SLD with vs. without liver fibrosis, whether these differences exist after considering the potential impact of confounding factors is unclear [25]. Indeed, considering the complex relationship between SLD/liver fibrosis, obesity, T2DM, and insulin resistance, it is challenging to establish clear and independent associations between specific BA concentrations and these individual diseases/conditions [9]. As such, we quantified circulating concentrations of 15 BA species in a highly characterised clinical cohort (INSYTE cohort [26,27]) consisting of 89 patients with SLD, with the aim of characterising differences in patients with and without clinically significant liver fibrosis (≥F2). Upon stratification of this cohort into those with (*n* = 30) or without (*n* = 59) clinically significant liver fibrosis (≥F2) (distinguished using a vibration-controlled transient elastography threshold of ≥8.2 kPa [28]), total BA concentrations were found to be higher in patients with SLD who also had ≥F2 compared to those with <F2 liver fibrosis (Figure 2A). Differences in total BAs between groups were largely driven by an increase in glycine- and taurine-conjugated primary and secondary BAs (Figure 2A). Of the 15 BA species quantified, concentrations of five glycine- or taurine-conjugated BAs (GCA, GCDCA, TCA, and TCDCA) were significantly higher (FDR < 0.05) in patients with ≥F2 liver fibrosis (Figure 2B,C).

To determine whether the associations between these five BA concentrations and the presence of ≥F2 liver fibrosis were independent of potential confounding factors, binary logistic regression analysis was performed where the absence vs. presence of ≥F2 liver fibrosis was the outcome variable. Univariate correlative analysis identified BMI, HOMA-IR, and interleukin (IL)-8 concentrations as potential confounding factors and these were included along with age, sex, and BA concentration as explanatory variables (Table 1). Accordingly, we found that only concentrations of GCA (Odds ratio (OR); 1.001, 95% confidence interval (CI); 1.000–1.001, *p* = 0.04), TCDCA (OR; 1.008, 95%CI; 1.000–1.015, *p* = 0.04), and GCDA (OR; 1.001, 95%CI 1.000–1.002, *p* = 0.04) remained positively yet weakly associated with the presence of ≥F2 liver fibrosis. Whilst in support of a connection between specific BA species and liver fibrosis in patients with SLD, these findings indicate that BA concentrations are also associated with other cardiometabolic risk factors that are also known to be associated with SLD severity. Therefore, whilst consistent evidence is emerging supporting a link between specific (largely conjugated) BAs and liver fibrosis severity in patients with SLD, it remains unclear whether these associations are independent of systemic metabolic dysfunction.

### 3.2. The Role of the Gut-Microbiota-Bile Acid Axis in MASLD Pathophysiology

Whilst a wealth of studies support the notion that BA concentrations and compositions are altered in individuals with MASLD (i.e., increased presence of conjugated BAs), the impact of these changes on MASLD pathophysiology remains unclear. Alterations in BAs in those with MASLD are likely driven by a range of mechanisms including alterations in the interactions between BA and the GM during the development of gut dysbiosis and in the handling of BAs within the liver [29]. For example, the presence of hepatic inflammation may downregulate hepatocyte BA uptake transporters (e.g., bile salt export pump [BSEP] and sodium taurocholate co-transporting polypeptide [NTCP]) leading to an increase in hepatic BA concentrations and BA leakage into portal blood [12] (Figure 3). The increased proportion of FXR antagonistic BAs (such as DCA) contributes to the suppression of hepatic FXR-, fibroblast growth receptor factor-4 and -5 (FGFR-4), and (FGFR-5) mediated signalling [30,31]. FXR signalling (both hepatic and intestinal) is a crucial regulator of hepatic lipid and glucose metabolism and has been extensively studied [32,33,34]. The suppression of FXR and Takeda G protein-coupled receptor 5 (TGR5)-signalling in MASLD leads to detrimental alterations in hepatic glucose and lipid homeostasis and the exacerbation of hepatic steatosis and systemic metabolic dysfunction (Figure 3) [34]. Intestinal and hepatic FXR activity is also paramount for the regulation of BA synthesis, and suppression of this signalling is thought to contribute to the elevated BA synthesis and altered enterohepatic circulation of BAs in individuals with MASLD [33]. The presence of hydrophobic BAs within the liver is thought to increase and activate resident hepatic immune cells and contribute to hepatic dysfunction and MASLD progression. Indeed, pre-clinical evidence indicates that modulating BA composition may be an effective approach to restoring FXR and TGR5-signalling—preventing the development of hepatic inflammation and associated metabolic disorders [31]. Moreover, increasing hyodeoxycholic acid (HCA) concentrations may alleviate SLD, suggesting that the regulation of specific BA concentrations may be an effective treatment strategy for SLD [35].

Additionally, other work directly implicated BAs in the pathophysiology of MASLD. In 2021, Xie and colleagues suggested that conjugated secondary 12α-hydroxylated BAs may contribute to liver fibrogenesis via the activation of hepatic stellate cells (HSCs) through TGR5/extracellular signal-regulated kinase 1/2 (ERK1/2) and TGR5/p38 mitogen-activated protein kinases (p38MAPK)-dependent mechanisms (Figure 3) [24]. Moreover, the depletion of *Tgr5* in hepatic stellate cells (HSCs) significantly decreased the development of liver fibrosis in a carbon tetrachloride-induced liver fibrosis mouse model [24]. However, it should be noted that the authors did not indicate whether HSC-specific depletion of *Tgr5* improved liver fibrosis severity in the other two (streptozocin-high-fat diet and long-term high-fat diet fed) murine models of liver dysfunction. Emerging evidence also highlights the potential importance of BAs in the peripheral circulation as a possible modulator of white adipose tissue function, which is at the nexus of metabolic homeostasis [36]. In this sense, various pre-clinical studies have suggested that specific BA species have different effects on the expression of inflammatory genes in adipocytes (as reviewed in [36]). Moreover, work from Velazquez-Villegas and colleagues indicates that TGR5 signalling promotes mitochondrial fission and the beiging of murine white adipose tissue supporting the role of TGR5 BA ligands as potential regulators of adipose tissue thermogenesis [37]. That said, it should be noted that investigations into the role of peripheral BAs on adipose tissue function are in their infancy, and fundamental questions regarding how BAs are handled and signal within this important tissue require addressing.

## 4. Association between Bile Acids and Cardiovascular and Cardiac Diseases

Whilst conflicting results exist between studies [38,39,40,41], some evidence indicates that BAs might be a relevant metabolite in relation to CVDs (Appendix A). Indeed, early work by Steiner and colleagues did not detect differences in circulatory BAs in patients with CAD compared to those without significant coronary artery disease (CAD) [38]. Conversely, more recent work suggested that total BAs [39,42] and primary, secondary, and conjugated BAs were all lower in patients with CAD compared to those without CAD [42]. These findings appear to be consistent with work by Feng and colleagues, which also suggests that total BA concentrations were negatively associated with the presence of CAD in post-menopausal women [41]. In contrast, evidence also supports a positive association between some BAs and CVDs. An example of such evidence can be seen in the work by Mayerhofer et al. in 2017 who explored circulating BA concentrations in 142 patients with heart failure and 20 sex- and age-matched healthy control participants [43]. The authors found that concentrations of various primary BAs decreased while concentrations of specific secondary BAs (namely GUDCA, GLCA, and TLCA) were higher in patients with heart failure compared to the healthy control participants [43]. More recently, higher TBA concentrations predicted major adverse cardiovascular events (MACEs) over a 2-year follow-up period in 425 patients with acute coronary syndrome (Hazard ratio; 1.03, 95%CI 1.01–1.05, *p* < 0.001) after adjusting for potential confounding factors [44]. Expanding on this work, Mateu-Fabregat and colleagues explored the concentrations of specific BAs in 309 patients with acute coronary syndrome and associations between baseline BA concentrations and MACEs over a mean follow-up period of 6.7 ± 3.6 years [45]. Accordingly, the authors identified that circulating concentrations of DCA, CDCA, GCA, and GUDCA were positively associated whilst CA, TDCA, GCDCA, hyodexoycholic acid (HDCA), UDCA, and LCA were negatively associated with the risk of MACE [45].

Further work is required to establish whether an association between total/specific BA species and the risk of CVDs exists and whether this association is independent of other potential confounding factors. Those studies employing approaches that facilitate the analysis of specific BAs (rather than total BAs) appear to suggest that not all BAs are equal in CVDs [43,45]. Indeed, this notion is somewhat consistent with observations made in relation to SLD (as discussed earlier) and could indicate a role of increased concentrations of specific (potentially secondary) BAs as a factor impacting cardiometabolic health. Moreover, whilst others have postulated that BAs may contribute to CVD in those with SLD [46], there is a scarcity of human data linking BAs to CVD risk in patients with SLD. Zhou et al. demonstrated in patients with MASLD that serum concentrations of UDCA and HCA were lower in patients with vs. without heart failure with a preserved ejection fraction [47]. Moreover, concentrations of TUDCA and GUDCA were lower in patients with MASLD and heart failure compared to those with just MASLD [47]. Whilst further work is required on this topic, these findings support the potential role of specific BAs in the development of CVD in patients with existing steatotic liver disease.

## 5. The Potential Role of BAs in Linking SLD and CVD and Cardiac Disease

As mentioned earlier, evidence indicates that the regulation of BAs and several BA signalling pathways is likely to become deranged in individuals with obesity-associated metabolic dysfunction and could have a profound impact on cardiometabolic risk [30,31]. The activation of FXR by BAs has a protective role on cardiometabolic health by regulating lipid and glucose metabolism, and also improves hepatic inflammation in murine models [32]. Consequently, it is plausible that derangements in FXR signalling (both hepatic and intestinal) associated with perturbed BA homeostasis in individuals with SLD exacerbate systemic metabolic dysfunction and contribute to CVD risk (Figure 3). In addition to FXR, recent work highlights the potential importance of TGR5 signalling as a modulator of CVD risk [48]. Specifically, dual ablation of FXR and TGR5 in mice promoted hypotension, aortic inflammation, intestinal dysbiosis, and BA synthesis dysregulation, indicating the relevance of these genes in maintaining intestinal, hepatic, and cardiovascular homeostasis [49]. Since both FXR and TGR5 signalling pathways are suppressed in murine models of obesity, it is possible that these derangements contribute to systemic metabolic dysfunction, MASLD progression, and CVD risk.

Along with the detrimental changes in the regulation of BA homeostasis and their downstream signalling typically occurring in those with obesity-associated metabolic dysfunction, it is also important to acknowledge the potential direct effects BAs have on cardiac health and function. Indeed, chronic elevations in BAs are directly cardiotoxic and lead to cardiomyopathy in murine models [50]. Moreover, murine models of BA overload displayed cardiac hypertrophy, bradycardia, and exercise intolerance, which were reversed upon BA sequestration [50]. Mao and colleagues identified that a carnitine acetyltransferase-mediated BA synthesis pathway interconnected cardiac energy metabolism, cholesterol homeostasis, and cardiomyocyte-intrinsic innate immune response and contributed to chronic myocardial inflammation and the progression of heart failure [51]. Moreover, this work suggested that an increased accumulation of the BA intermediate 7α-hydroxyl-3-oxo-4-cholestenoic acid (7-HOCA) within cardiomyocytes induced mitochondrial stress and cellular inflammation, which contributes to heart failure (Figure 3) [51]. BAs have also been shown to directly modulate cardiovascular function via activating a host of BA receptors including FXR, PXR, TGR5, and the vitamin D receptor (VDR) [52,53]. The consequences of specific BA receptor activation within the cardiovascular system are also thought to be cell type- and dose-dependent, adding additional complexity to the pursuit by researchers to understand the relevance of BA signalling in CVD. Whilst ongoing, research indicates that FXR and TGR5 signalling within the cardiovascular system appears to be largely cardioprotective and anti-atherogenic [54]. Indeed, whilst a detailed discussion of BA signalling within the cardiovascular system is beyond the scope of this review, other recent reviews have provided detailed discussions of this topic [54,55].

That said, it is important to emphasise that the modulation of FXR signalling in the context of cardiovascular disease might be a double-edged sword. In 2015, the FXR agonist obeticholic acid (OCA) improved liver histology outcomes in patients with non-alcoholic steatohepatitis in a randomised, placebo-controlled clinical trial [56]. In this trial, OCA treatment was also associated with an increase in total and low-density lipoprotein (LDL) cholesterol concentrations compared to the placebo control, which was attenuated after treatment was stopped. Other work has since confirmed this effect of OCA on pro-atherogenic LDL production, raising concerns surrounding the long-term agonism of FXR receptors for treating cardiometabolic diseases [57,58]. The impact of OCA on LDL concentrations is thought to result largely from FXR-mediated repression of BA synthesis, which consequently drives hepatic cholesterol accumulation. Subsequently, cholesterol-rich LDL production is increased, potentially exacerbating the pro-atherogenic lipid phenotype that commonly features in individuals with metabolic dysfunction and SLD. Thus, whilst short-term FXR agonism appears to be cardiometabolically protective, concerns exist surrounding long-term FXR activation on CVD risk. In the event of OCA being approved as an SLD therapy, additional lipid-lowering treatment will most likely need to be prescribed to mitigate rises in LDL concentrations [58]. Indeed, next, we will briefly discuss the therapeutic potential of BAs and BA receptor modulators to tackle SLD and CVD and provide our thoughts on the best therapeutic combination partner for OCA.

## 6. Therapeutic Potential of BAs and BA Receptor Modulators to Tackle SLD and CVD

Recent progress in drug discovery for MASH and liver fibrosis has led to the Food and Drug Administration’s approval of Resmetirom [59]. This approval provides hope for more effective management of MASH, underscoring the need for continued innovation and multi-faceted treatment strategies. Resmetirom is a thyroid hormone receptor beta (THRβ) agonist, which is highly expressed in hepatocytes and mediates the effects of triiodothyronine (T3) on metabolic control [60]. It is also potentially important in the context of MASLD as a multisystem disease that Resmetirom has beneficial effects on lipoprotein metabolism, and Resmetirom may therefore have an additional beneficial effect to ameliorate CVD risk in patients with MASLD [61]. THRβ activation helps maintain hepatic lipid homeostasis by enhancing intrahepatic lipolysis and fatty acid oxidation, improving mitochondrial function, and stimulating cholesterol turnover [60,62]. Additionally, THRβ activation suppresses hepatic CYP8B1 expression (Figure 1), leading to reduced production of 12α-hydroxylated bile acids (such as TCA) and subsequently decreased intestinal lipid absorption, contributing to the improvement of NASH [63]. Resmetirom’s impact on bile acid homeostasis is significant, altering bile acid composition by reducing 12α-hydroxylated bile acid content and the ratio of 12α-hydroxylated to non-12α-hydroxylated bile acids [63].

In addition to Resmetirom, other therapeutic strategies have shown promise in the management of NASH and also have important effects on BA metabolism. FXR agonists, such as OCA [56], and non-steroidal alternatives like Tropifexor and GS-9674 [64] have shown efficacy in reducing liver fat and improving liver function, though side effects like pruritus and LDL-cholesterol increases remain a concern (Figure 1). TGR5 agonists, like INT767, have demonstrated potential in reducing hepatic steatosis and inflammation [65]. Additionally, bile acid sequestrants and compounds like Aldafermin (a fibroblast growth factor-19 (FGF19) analogue) [66] and HTD1801 [67] (a combination of berberine and UDCA) have shown promise in clinical trials [68].

Given the complex and multifactorial nature of SLD and MASH, it is unlikely that a single medication will address all aspects of the disease. A multi-targeted therapeutic approach is suggested, combining FXR agonists with lipid regulators, insulin sensitizers, or GLP-1 analogues to manage lipid and glucose metabolism. The combination of molecules with different modes of action may offer better benefits, particularly in patients with additional complications. Additionally, classifying patient subsets based on biomarker screening could optimize treatment strategies, addressing the heterogeneous nature of MASH and its associated systemic metabolic abnormalities.

## 7. Future Perspectives and Conclusions

There have been great advances in our understanding of the homeostatic importance and potential pathophysiological roles of BAs in MASLD and CVDs in recent years. Future work should look to assess the potential use of specific BA concentrations for the prediction of CVD risk in populations with MASLD. Moreover, given the heterogeneous nature of MASLD, additional work is required to explore the relationships between TBA and individual BA concentrations with MASLD severity. Whilst emerging evidence highlights the potentially important role of BA-mediated modulation of adipose tissue function and inflammation, there are still many unanswered research questions surrounding this area. It is apparent that aberration from physiological hepatic BA signalling may exacerbate MASLD severity; however, whether derangements in BA signalling in adipose tissue contribute to adipose tissue dysfunction in individuals living with metabolically unhealthy obesity is unclear. Further randomised placebo-controlled clinical trials are required to identify treatment combinations to mitigate the LDL-increasing effects of OCA and test the efficacy of these combinations to alleviate MASLD and CVD severity. Collectively, a wealth of evidence now supports the role of BAs and BA signalling in the development of progression of both MASLD and CVD and cardiac disease. The potential to combine new and approved MASLD treatments with BA receptor-modulating compounds provides an exciting potential therapeutic strategy for patients impacted by MASLD and CVD.

## Figures and Tables

**Figure 1 biomolecules-14-00841-f001:**
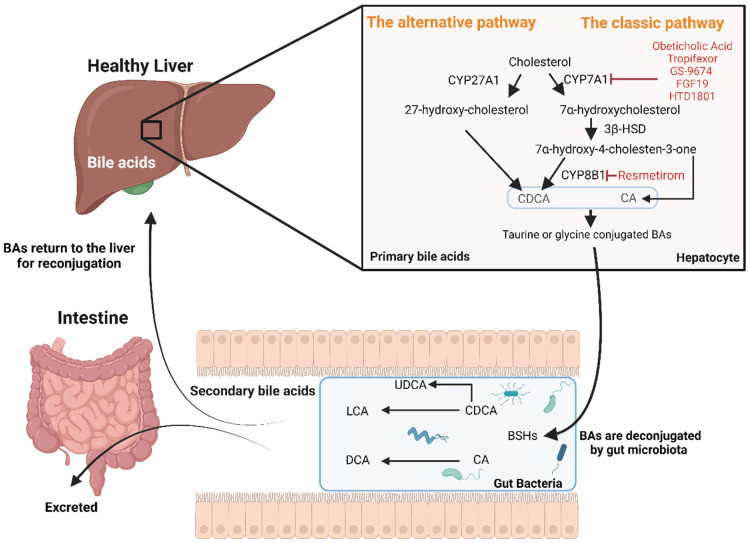
Metabolism of cholesterol and bile acid synthesis and newer medications affecting bile acid metabolism. The liver is the primary site for bile acid synthesis, occurring through two main pathways: the classical and alternative. In the classical pathway, cholesterol is converted into CDCA and CA through a series of enzyme-mediated steps, including the action of CYP7A1 and CYP8B1. In the alternative pathway, cholesterol undergoes transformation via different enzymes such as CYP27A1. Following synthesis, primary bile acids are conjugated with taurine or glycine, enhancing their hydrophilicity before secretion into the bile canaliculi. Once in the intestine, bile acids are deconjugated by BSHs to form unconjugated bile acids. These are further processed where CA is converted to DCA and CDCA to LCA, thus reducing their solubility. Additionally, a small portion of CDCA is transformed by gut microbiota into UDCA. These transformations result in secondary bile acids, which are either reabsorbed and returned to the liver or excreted in the faeces. Newer medications affecting bile acid metabolism, highlighted in red font in the figure, include Resmetirom, FXR agonists (Obeticholic acid, Tropifexor, and GS-9674), and compounds like the FGF19 analogue (Aldafermin and HTD1801). These agents have been explored for their potential to reduce liver fat, improve liver function, and modulate inflammation in the management of MASH. Created using BioRender.com. Abbreviations: BA; bile acid, CDCA; chenodeoxycholic acid, CA; cholic acid, CYP7A1; cholesterol 7α-hydroxylase, CYP8B1; sterol 12α-hydroxylase, CYP27A1 sterol 27-hydroxylase, BSHs; bile salt hydrolases, DCA; deoxycholic acid, LCA; lithocholic acid, UDCA; ursodeoxycholic acid, FXR; farnesoid X receptor, FGF19; fibroblast growth factor-19, MASH; metabolic-dysfunction associates steatohepatitis.

**Figure 2 biomolecules-14-00841-f002:**
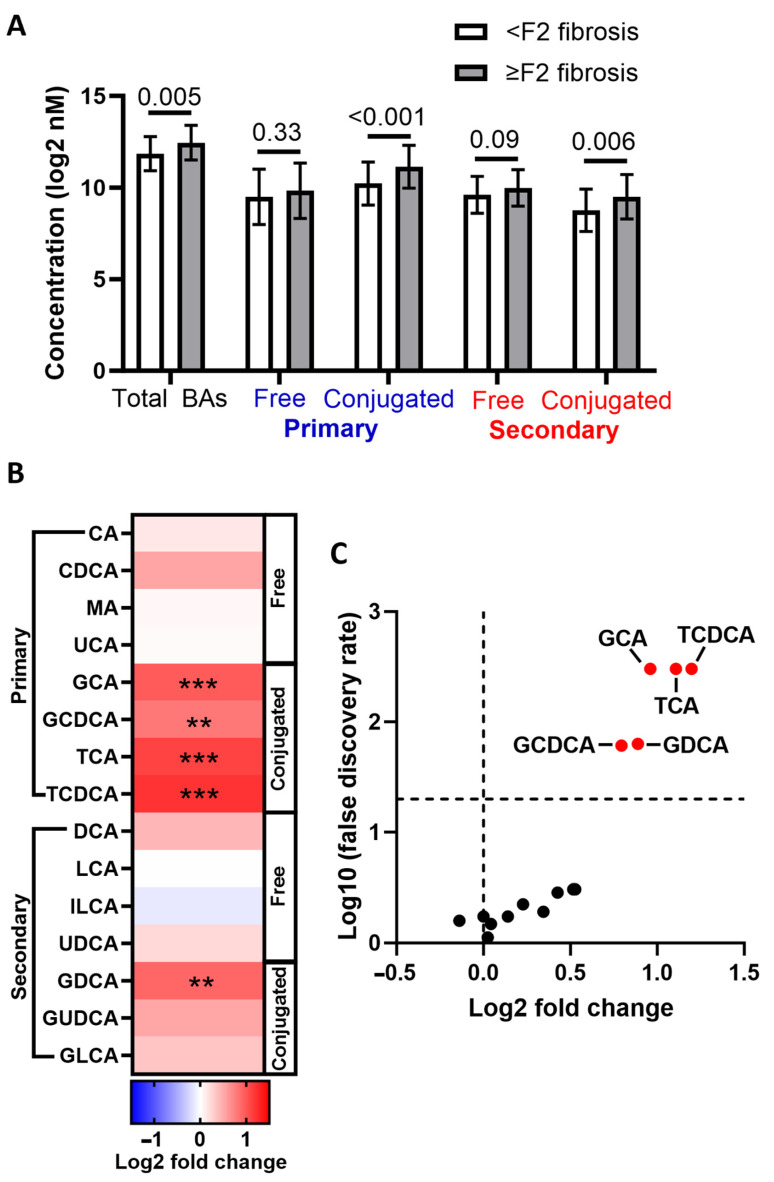
Serum concentrations of several conjugated bile acids are elevated in patients with SLD with vs. without ≥F2 liver fibrosis. (**A**) Comparison of total, free, and conjugated primary and secondary BAs in patients without (*n* = 59) vs. with (*n* = 30) ≥F2 liver fibrosis. (**B**) Heatmap showing differences in specific BA concentrations (nM) in patients with vs. without ≥F2 liver fibrosis. The *y*-axis denotes the bile acid. Colour coding depicts log2 fold change in patients with compared to those without ≥F2 liver fibrosis. The brighter the red colour, the more increased the BAs were in patients with ≥F2 liver fibrosis. (**C**) Volcano plot showing differences in serum BA concentrations (nM) in patients with vs. without ≥F2 liver fibrosis. The *y*-axis denotes the log10 *q*-value of an unpaired two-sample Student’s *t*-test comparing BA concentrations after adjusting for multiple testing using the Benjamini–Hochberg method (FDR threshold of 0.05). Red dots denote significantly increased BAs in patients with ≥F2 liver fibrosis and are labelled. ** *p* < 0.001, *** *p* < 0.0001. Abbreviations: BA; bile acid, CA; cholic acid, CDCA; chenodeoxycholic acid, MA; Muricholic acid, UCA; ursocholic acid, GCA; glycocholic acid, GCDCA; glycochenodeoxycholic acid, TCA; taurocholic acid, TCDCA; taurochenodeoxycholic acid, DCA; deoxycholic acid, LCA; litocholic acid, ILCA; isolithocholic acid, UDCA; ursodeoxycholic acid, GDCA; glycodeoxycholic acid, GUDCA; glycoursodeoxycholic acid, GLCA; glycolithocholic acid.

**Figure 3 biomolecules-14-00841-f003:**
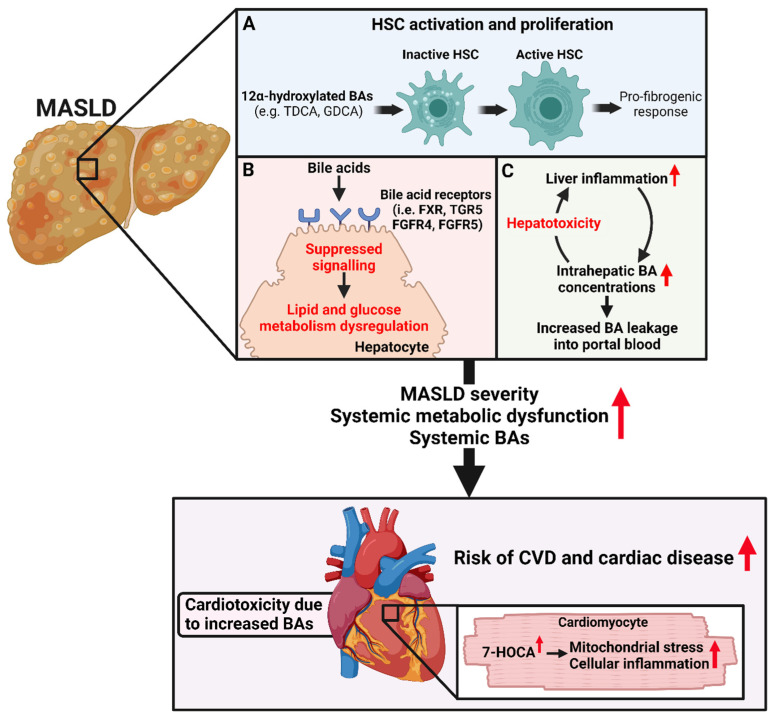
Putative roles of BAs in the pathophysiology of MASLD, CVD, and cardiac disease. In the liver, (**A**) 12α-hydroxylated BAs such as TDCA and GDCA have been found to induce the activation and proliferation of hepatic stellate cells and contribute to a pro-fibrogenic environment. (**B**) Derangements in hepatic BA signalling are thought to contribute to local and systemic perturbations in glucose and lipid metabolism, thus increasing cardiometabolic risk. (**C**) The presence of hepatic inflammation is thought to contribute to the impairment of hepatic BA metabolism and synthesis, leading to an increase in intrahepatic BA concentrations. These increases, especially in hydrophobic BA species such as DCA, are known to be hepatotoxic and further contribute to liver inflammation. Moreover, these increases in hepatic BA content are thought to also increase the leakage of BAs into portal blood. These hepatic effects resulting from the dysregulation of hepatic BA metabolism and signalling may exacerbate MASLD severity, systemic metabolic dysfunction, and increase systemic BA concentrations. Collectively, these factors may increase the risk of CVD and cardiac disease in patients with MASLD. Moreover, excess cardiac BAs are thought to be cardiotoxic and contribute to cardiac dysfunction. For example, increased presence of 7-HOCA in cardiomyocytes has been shown to induce mitochondrial stress and cellular inflammation. It should also be noted that derangements in BA metabolism and signalling in other tissues including the gastrointestinal system and adipose tissue are also likely to have an important role in driving MASLD, CVD, and cardiac disease risk. Black arrows indicate direction, red arrows indicate an increase. Created using BioRender.com. Abbreviations: MASLD; metabolic-dysfunction associated steatotic liver disease, BA; bile acid, TDCA; taurodeoxycholic acid, GDCA; glycodeoxycholic acid, HSC; hepatic stellate cell, FXR; farnesoid X receptor, TGR5; Takeda G protein-coupled receptor 5, FGFR4/5; fibroblast growth receptor factor 4/5, CVD; cardiovascular disease, 7-HOCA; 7α-hydroxyl-3-oxo-4-cholestenoic acid.

**Table 1 biomolecules-14-00841-t001:** Univariate associations with serum concentrations of BAs found to be different between patients with SLD with or without ≥F2 liver fibrosis.

Variable	GCA	GCDCA	TCA	TCDCA	GDCA
Age (years)	−0.07	−0.07	−0.09	−0.08	0.10
Systolic blood pressure (mmHg)	−0.12	−0.13	−0.05	−0.07	0.03
Diastolic blood pressure (mmHg)	−0.11	−0.10	−0.01	−0.05	−0.10
BMI (kg/m^2^)	0.37 **	0.32 **	0.26 *	0.26 *	0.24 *
DEXA lean body mass (kg)	0.05	0.02	0.05	0.02	0.03
DEXA total body fat (%)	0.23 *	0.20 *	0.11	0.11	0.19
MRI SAT (%) †	0.22 *	0.15	0.14	0.12	0.18
MRI VAT (%) †	−0.03	0.001	−0.08	−0.08	0.05
Fasting glucose (mmol/L) †	0.12	0.14	0.07	0.03	0.19
Haemoglobin A1c (mmol/mol) †	0.12	0.11	0.05	0.01	0.20
Fasting insulin (mIU/L) †	0.38 **	0.36 **	0.39 **	0.42 ***	0.23 *
HOMA-IR †	0.45 ***	0.41 ***	0.45 ***	0.42 ***	0.36 **
AdipoIR †	0.21	0.18	0.22 *	0.25 *	0.18
Triglycerides (mmol/L) †	0.18	0.15	0.21 *	0.17	0.24 *
Total cholesterol (mmol/L) †	0.04	0.01	0.08	0.11	−0.02
HDL cholesterol (mmol/L)	−0.09	−0.08	−0.05	0.001	−0.01
AST (IU/L) †	0.31 *	0.23 *	0.33 **	0.33 **	0.29 *
ALT (IU/L) †	0.24 *	0.15	0.30 *	0.27 *	0.24 *
MRS-measured liver fat (%) †	0.15	0.09	0.21 *	0.20 *	0.23 *
Liver VCTE (kPa) †	0.39 ***	0.32 *	0.35 **	0.36 **	0.34 **
FIB-4 index †	0.21 *	0.09	0.16	0.17	0.26 *
ELF test †	0.09	0.05	0.11	0.12	0.16
APRI index †	0.26 *	0.14	0.28 *	0.25	0.25 *
Adiponectin (μg/mL) †	−0.19	−0.13	−0.25	−0.18	−0.06
Leptin (ng/mL) †	0.26 *	0.20	0.16	0.16	0.25 *
TNFα (pg/mL) †	0.29 *	0.23 *	0.33 *	0.35 **	0.18
IL-6 (pg/mL) †	0.36 **	0.26 *	0.33 *	0.33 **	0.18
IL-8 (pg/mL) †	0.42 ***	0.308	0.37 **	0.32 *	0.35 **
IL-10 (pg/mL) †	0.28 *	0.16	0.31 *	0.30 *	0.17
hs-CRP (mg/L) †	0.12	0.17	0.13	0.22 *	0.03
GDF-15 (pg/mL)	0.24 *	0.19	0.18	0.14	0.21 *

Correlations were performed using Pearson’s or Spearman’s (†) coefficients for normally and non-normally distributed data, respectively. * *p* < 0.05, ** *p* < 0.001, *** *p* < 0.0001. Abbreviations: BMI; body mass index, DEXA; dual-energy X-ray absorptiometry, HOMA-IR homeostatic model assessment for insulin resistance, AdipoIR; adipose tissue insulin resistance index, MRI; magnetic resonance imaging, SAT; subcutaneous adipose tissue, VAT, visceral adipose tissue, HDL; high-density lipoprotein, AST; aspartate aminotransferase, ALT; alanine transaminase, MRS; magnetic resonance spectroscopy, 13C-KICA BT; 13C-ketoisocaproate breath test, VCTE; vibration-controlled transient elastography, APRI; AST to platelet ratio index, FIB-4; Fibrosis-4, ELF; enhanced liver fibrosis, GDF-15; growth differentiation factor-15, TNFα; tumour necrosis factor-α, IL; interleukin, hs-CRP; high-sensitivity C-reactive protein.

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
