# Peer review of "Bile Acids as Emerging Players at the Intersection of Steatotic Liver Disease and Cardiovascular Diseases"

_biomolecules, 2024, doi:10.3390/biom14070841_

Round 1

Reviewer 1 Report

Comments and Suggestions for Authors

This is an interesting review, which summarizes the association of bile acids in metabolic patterns, cardiovascular diseases and MASLD. Also, there is a need for more work in this field for a more comprehensive understanding of the role of bile acids. However, I must say that the title it is not very plain. The title must be "Bile Acids as Emerging Players at the Intersection of Steatotic Liver Disease and Cardiovascular diseases"

Comments on the Quality of English Language

There are very few typos like some of the miss commas before "and". 

Author Response

Comment 1: This is an interesting review, which summarizes the association of bile acids in metabolic patterns, cardiovascular diseases and MASLD. Also, there is a need for more work in this field for a more comprehensive understanding of the role of bile acids. However, I must say that the title it is not very plain. The title must be "Bile Acids as Emerging Players at the Intersection of Steatotic Liver Disease and Cardiovascular diseases"

Response 1: We thank the Reviewer for their support and kind words. As per the Reviewer's suggestion, the title of the manuscript has now been changed to "Bile Acids as Emerging Players at the Intersection of Steatotic Liver Disease and Cardiovascular diseases"

Comment 2:  There are very few typos like some of the miss commas before "and". 

Response 2: We thank the Reviewer for noticing these oversights and we have been through the manuscript carefully and made spelling and grammatical corrections.

Reviewer 2 Report

Comments and Suggestions for Authors

The article "Bile Acids as Emerging Players at the Intersection of Steatotic Liver Disease and Cardiovascular and Cardiac diseases" focuses on the evolving understanding of bile acids (BAs) as central players linking steatotic liver disease (SLD) with cardiovascular and cardiac diseases. Affecting approximately 25% of the world's population, steatotic liver diseases are a serious public health problem and are associated with obesity, dyslipidemia, and insulin resistance. These conditions increase cardiovascular risk. Bile acids, in addition to their role in the digestion of fats, act as signaling molecules that regulate glucose, lipids and the balance of intestinal microflora. Bile acids have been recognized as critical mediators of cardiovascular health, influencing vascular tone, cholesterol homeostasis, and inflammatory responses. Altering the characteristics of bile acids may offer new perspectives for the treatment not only of liver diseases but also of cardiovascular diseases associated with metabolic dysfunctions.

The article summarizes the evidence on the interconnection of bile acids and cardiovascular diseases in the cases of metabolic dysfunctions and steatotic liver diseases. It provides a comprehensive analysis of the mechanisms through which bile acids affect cardiovascular health and highlights the need for further research to understand the precise effects of bile acids and their receptors on these diseases.

In my opinion it deserves to be published in its present form.

Author Response

Comment 1: 

The article "Bile Acids as Emerging Players at the Intersection of Steatotic Liver Disease and Cardiovascular and Cardiac diseases" focuses on the evolving understanding of bile acids (BAs) as central players linking steatotic liver disease (SLD) with cardiovascular and cardiac diseases. Affecting approximately 25% of the world's population, steatotic liver diseases are a serious public health problem and are associated with obesity, dyslipidemia, and insulin resistance. These conditions increase cardiovascular risk. Bile acids, in addition to their role in the digestion of fats, act as signaling molecules that regulate glucose, lipids and the balance of intestinal microflora. Bile acids have been recognized as critical mediators of cardiovascular health, influencing vascular tone, cholesterol homeostasis, and inflammatory responses. Altering the characteristics of bile acids may offer new perspectives for the treatment not only of liver diseases but also of cardiovascular diseases associated with metabolic dysfunctions.

The article summarizes the evidence on the interconnection of bile acids and cardiovascular diseases in the cases of metabolic dysfunctions and steatotic liver diseases. It provides a comprehensive analysis of the mechanisms through which bile acids affect cardiovascular health and highlights the need for further research to understand the precise effects of bile acids and their receptors on these diseases.

In my opinion it deserves to be published in its present form.

Response 1: 

We are very grateful for the Reviewer's kind words and are happy they found our manuscript interesting.